# Dynamic Survival Risk Prognostic Model and Genomic Landscape for Atypical Teratoid/Rhabdoid Tumors: A Population-Based, Real-World Study

**DOI:** 10.3390/cancers16051059

**Published:** 2024-03-05

**Authors:** Sihao Chen, Yi He, Jiao Liu, Ruixin Wu, Menglei Wang, Aishun Jin

**Affiliations:** 1Department of Immunology, School of Basic Medical Sciences, Chongqing Medical University, Chongqing 400010, China; chensh950925@foxmail.com (S.C.); h1512370962@foxmail.com (Y.H.); 191093@hospital.cqmu.edu.cn (R.W.); 2Chongqing Key Laboratory of Tumor Immune Regulation and Immune Intervention, Chongqing 400010, China; 3Children’s Hospital of Chongqing Medical University, Chongqing 400010, China; liujiao6969@foxmail.com; 4Department of Pediatrics, Women and Children’s Hospital of Chongqing Medical University, Chongqing 400010, China; 5Department of Pediatrics, Chongqing Health Center for Women and Children, Chongqing 400010, China

**Keywords:** atypical teratoid/rhabdoid tumor, LASSO, Random Forest, SEER, COSMIC, risk prognostic model, Genomic Landscape

## Abstract

**Simple Summary:**

An atypical teratoid/rhabdoid tumor (AT/RT) is an uncommon, yet aggressive, pediatric central nervous system neoplasm. Our prognostic study included 316 Surveillance, Epidemiology, and End Results (SEER) repository participants and 27 external validation patients. The incidence of AT/RT consistently increased between 2000 and 2020. Age, SEER stage, tumor size, surgery, chemotherapy, and radiotherapy are closely related to the prognosis of AT/RT. Triple therapy resulted in discernibly enhanced OS and CSS. The most common mutations in AT/RT identified using the COSMIC database were SMARCB1, BRAF, SMARCA4, NF2, and NRAS. Our study identified the clinical determinants of prognosis in patients with AT/RT and mapped the genetic mutation landscape. The prediction model that we devised may offer a valuable tool to address existing clinical challenges. Additionally, analysis based on mutational genomics will facilitate the research regarding molecular-targeted drugs.

**Abstract:**

Background: An atypical teratoid/rhabdoid tumor (AT/RT) is an uncommon and aggressive pediatric central nervous system neoplasm. However, a universal clinical consensus or reliable prognostic evaluation system for this malignancy is lacking. Our study aimed to develop a risk model based on comprehensive clinical data to assist in clinical decision-making. Methods: We conducted a retrospective study by examining data from the Surveillance, Epidemiology, and End Results (SEER) repository, spanning 2000 to 2019. The external validation cohort was sourced from the Children’s Hospital Affiliated to Chongqing Medical University, China. To discern independent factors affecting overall survival (OS) and cancer-specific survival (CSS), we applied Least Absolute Shrinkage and Selection Operator (LASSO) and Random Forest (RF) regression analyses. Based on these factors, we structured nomogram survival predictions and initiated a dynamic online risk-evaluation system. To contrast survival outcomes among diverse treatments, we used propensity score matching (PSM) methodology. Molecular data with the most common mutations in AT/RT were extracted from the Catalogue of Somatic Mutations in Cancer (COSMIC) database. Results: The annual incidence of AT/RT showed an increasing trend (APC, 2.86%; 95% CI:0.75–5.01). Our prognostic study included 316 SEER database participants and 27 external validation patients. The entire group had a median OS of 18 months (range 11.5 to 24 months) and median CSS of 21 months (range 11.7 to 29.2). Evaluations involving C-statistics, DCA, and ROC analysis underscored the distinctive capabilities of our prediction model. An analysis via PSM highlighted that individuals undergoing triple therapy (integrating surgery, radiotherapy, and chemotherapy) had discernibly enhanced OS and CSS. The most common mutations of AT/RT identified in the COSMIC database were SMARCB1, BRAF, SMARCA4, NF2, and NRAS. Conclusions: In this study, we devised a predictive model that effectively gauges the prognosis of AT/RT and briefly analyzed its genomic features, which might offer a valuable tool to address existing clinical challenges.

## 1. Introduction

An atypical teratoid/rhabdoid tumor, commonly referred to as AT/RT, is an exceptionally aggressive form of central nervous system (CNS) tumor known for its prognostic outlook [1,2]. Notably, it predominantly targets children below the age of three years, constituting less than 5% of all pediatric CNS tumors. However, this percentage rises to 20% when focusing solely on the subgroup of children under the age of three [3]. Although there are some histopathological similarities between AT/RT and other embryonal tumors of the CNS (such as, medulloblastoma, neuroblastoma), it was only in 1996 that AT/RT was acknowledged as a unique tumor [4]. In 2016, the World Health Organization (WHO) categorized it as a grade IV embryonal malignancy of the CNS [5]. Delving into its genetic underpinnings, a significant number of affected children exhibit mutations in genes associated with chromatin alterations. This includes, but is not limited to, the SMARCB1 (INI-1) gene on chromosome 22q11.2 and the SMARCA4 (BRG1) gene on chromosome 19p13.2 [6,7]. These genetic changes are pivotal for diagnostic assessment of AT/RT. Clinically, patients often experience symptoms such as vomiting, gait imbalance, and recurrent seizures, with the disease advancing at an alarming rate [4]. The median survival duration typically hovers around a mere year [8].

The infrequency of AT/RT in the general population and its diagnostic intricacies mean that prior investigations of this ailment have largely focused on individual case analyses and modest retrospective evaluations [9]. There is an existing void concerning a robust prognostic staging framework and authoritative guidance for the most effective therapeutic approaches. Recent studies have underscored that conventional treatments, including surgical interventions, chemotherapy, and radiation therapy, continue to be the predominant therapeutic choices, even though their survival enhancements remain limited [10]. An appraisal of prolonged survival data from extant cases underlines the effectiveness of a tripartite approach, amalgamating surgical procedures with chemotherapy and radiation therapy, in increasing patient longevity [11]. Additionally, post-surgical localized radiation combined with systemic chemotherapy is indispensable, considering the pronounced invasiveness and propensity for metastasis of the tumor [12]. Evidence suggests that high-dose alkylating agent chemotherapy, along with intrathecal chemotherapy, constitutes potent systemic therapeutic avenues, markedly elevating survival rates in pediatric cohorts [13]. While radiation offers amplified tumor containment postoperatively, its repercussions on neural development in children, potentially instigating enduring neurocognitive impairments, spur contention in its clinical application [14].

Amid prevailing clinical uncertainties, our research endeavors to conceive and corroborate a dynamic assessment tool for survival risks. This tool draws upon comprehensive demographic data and integrates clinical and genomic characteristics. The objective is not only to surpass the shortcomings intrinsic to current prognostic frameworks but also to enrich the foundation upon which clinicians base their decisions.

## 2. Materials and Methods

### 2.1. Study Design and Selection Criteria

This study adhered to the Transparent Reporting of a Multivariable Prediction Model for Individual Prognosis or Diagnosis (TRIPOD) reporting guidelines for prognostic studies [15]. A comprehensive workflow is shown in Figure 1. Data retrieval was facilitated using the SEER* Stat software (version 8.4.1), accessing the most updated iteration of the SEER database.

Incidence data were acquired using the Incidence-SEER 22 Regs Research Limited-Field Data, Nov 2022 Sub (2000–2020), and the incidence rates were adjusted relative to the age of the standard American population as of 2000. Complete follow-up and treatment data were collected from the Incidence-SEER 17 Regs Research Plus Data, Nov 2021 Sub (2000–2019), and the criteria employed during the screening phase were detailed as follows. First, we only considered patients identified with an AT/RT diagnosis, bearing the International Classification of Diseases for Oncology, Third Edition (ICD-O-3) code 9508/3, between the years 2000 and 2019. The patient’s medical record should indicate the sole primary tumor. Second, the documented survival duration for patients was at least 1 month. Third, the dataset for each patient should encompass comprehensive follow-up data. Lastly, patient data had to incorporate critical information elements: vital status, duration of survival, demographics (including age, sex, and race), combined summary stage, CS tumor dimensions, and primary therapeutic interventions. For external validation, we enrolled 27 patients with AT/RT treated at the Children’s Hospital Affiliated to Chongqing Medical University from January 2018 to October 2023. Molecular data with common mutations in AT/RT were extracted from the Catalogue of Somatic Mutations in the Cancer (COSMIC) database. STRING database was used to collect and integrate potential protein interactions.

### 2.2. Statistical Analysis

Age-adjusted incidence rates were computed as per 10,000 individuals using the SEER statistic, and annual percentage changes (APCs) were also determined. Data sourced from the SEER database were randomly divided into two distinct sets: a training cohort and a validation cohort, at proportions of 70% and 30%, respectively. Categorical variables were assessed by tabulating their frequencies and expressed as percentages, and the chi-square test was then applied for their evaluation. To map out survival trends, the Kaplan–Meier technique was used, and any disparities among these curves were discerned using the log-rank test. To identify factors that had a significant influence on overall survival (OS) and cancer-specific survival (CSS), our approach focused on applying Least Absolute Shrinkage and Selection Operator (LASSO) and Random Forest (RF) regression analysis.

To assess model discrimination, we evaluated the area under the time-dependent ROC and utilized the C-index. Calibration plots were designed to compare the predicted survival rates with actual outcomes. To ascertain the predictive power of our system against SEER stage, we relied on both DCA and time-dependent ROC. Individualized risk scores were determined by leveraging the formulated nomograms. This led to the categorization of patients into groups with higher or lower risks using the Surv_Cutpoint function to identify the best cut-off values for OS and CSS. Visual heatmaps highlighted the associations between risk factors and spread of clinical features across different risk categories for OS and CSS. Sankey diagrams were crafted for every variable within the concluding risk category, thereby enriching the clinical applicability of our framework. To ensure a meticulous comparison of survival rates across various treatments, we integrated Propensity Score Matching analysis with a match tolerance/caliper of 0.02. The top 20 mutated genes derived from the COSMIC database were utilized for subsequent potential protein interaction network analysis (Confidence score > 0.7) and imported into Cytoscape software (v3.8.2) for visualization. For biological process and pathway enrichment analyses, the Kyoto Encyclopedia of Genes and Genomes (KEGG) and Gene Ontology (GO) analyses were performed using the R clusterProfiler package. Our analytical methods hinged on SPSS 26.0 and R software (version 4.1.1); all findings were deemed significant at *p* values less than 0.05.

## 3. Results

### 3.1. Epidemiological Characteristics Analysis

The incidence of AT/RT consistently increased between 2000 and 2020, with an APC of 2.86% (95% CI:0.75–5.01; *p* < 0.05) (Figure 2A). Distribution analysis showed that, regardless of gender differences, children younger than three years old accounted for the vast majority of the population (Figure 2B).

### 3.2. Clinical Characteristics of Patients

Our study incorporated data from 316 individuals diagnosed with AT/RT gathered from the SEER 17 Regs Research Plus database between 2000 and 2019. For analytical purposes, this patient cohort was divided in a 7:3 ratio, assigning 221 individuals to the training set and the remaining 95 to the validation set. We assessed the clinical characteristics to discern any potential disparities between the two subsets. Notably, the distribution did not indicate any marked discrepancies (*p* > 0.05) in demographic or clinical factors. Table 1 presents the demographic and clinical data. Key observations include the fact that a significant proportion of the study participants were infants aged under three years (*n* = 248, 78.5%), predominantly of Caucasian descent (*n* = 241, 76.3%), having primary tumors located intracranially (*n* = 299, 94.6%), and hailing from households with lower incomes (*n* = 221, 69.9%). In terms of disease progression, as classified by the SEER system, most patients presented with localized tumors at diagnosis (*n* = 189, 59.8%), followed by those with regional (*n* = 61, 19.3%) and metastatic disease (*n* = 66, 20.9%). Treatment modalities revealed that gross total resection/subtotal resection (GTR/STR) was performed in 94.0% of the cases, chemotherapy in 81.0%, and radiotherapy in 46.8%. The median survival span for all patients in the database was 18 months (range: 11.5–24.5), with a median cancer-specific survival of 21 months (range: 11.7–29.2). The training set exhibited a median OS of 19 months (range: 12.3–25.6) and CSS of 22 months (range: 12.6–31.3). For the validation set, these measures were 17 months (range: 10.8–23.2) and 20 months (range: 9.0–30.1), respectively. Moreover, 27 patients with AT/RT treated at the Children’s Hospital Affiliated to Chongqing Medical University were included in the external validation. This external set had a median OS of 10 months (range:5.8–14.2), and demographic and clinical details are provided in Appendix A.

### 3.3. Prognostic Factor Selection and Model Construction

Before delving into machine learning algorithm screening, we first evaluated the potential collinearity between all scrutinized parameters using Spearman correlation analysis, as shown in Figure 3A. To identify the best coefficient for each prognostic determinant, we employed the LASSO and RF algorithms, which ensured the circumvention of overfitting during the selection of important variables [16]. LASSO regression was performed by minimizing the partial probability deviation and generating coefficient curves from a logarithmic (lambda) series (Figure 3C,E). Guided by the requisite standards for Lasso–Cox regression and adopting a 10-fold cross-validation, the algorithm discerned six pivotal clinical parameters (age, SEER stage, tumor size, surgical interventions, chemotherapeutic approaches, and radiological treatments) with significance as standalone predictors in both the OS and CSS frameworks (Figure 3B,D). In the RF algorithms, by increasing the number of random forests, the out-of-bag error rate gradually decreases (Figure 3G,I), allowing for the determination of the importance index of each parameter in both OS and CSS (Figure 3H,J). Furthermore, the TOP 6 common parameters identified by both algorithms (Figure 3F) were selected as the final predictor variables for the model. Finally, we synthesized these selected prognostic markers into forest diagrams (Figure 4A,B) to devise nomogram-driven prognostic models for OS and CSS (Figure 4C,D).

### 3.4. Dynamic Web Version Survival Model

To support researchers and clinicians, our team launched digital versions of our nomograms designed to assess OS and CSS in patients with AT/RT. These tools are accessible at the following URLs: https://atrtapp.shinyapps.io/shinyNomoforATRTinOS/ and https://atrtapp.shinyapps.io/shinyNomoforATRTinCSS/ (accessed on 8 January 2024) (Figure 4E,F).

### 3.5. Internal and External Multidimensional Validation of Models

The nomogram showed notable capabilities in forecasting OS for intervals of one, two, and three years. Both the training (0.815) and validation (0.801) cohorts registered C-index values that outshone those of the SEER stage method, which scored 0.648 and 0.656, respectively. Furthermore, when considering one-, two-, and three-year CSS projections, our model surpassed the SEER-stage method, yielding C-index scores of 0.809 and 0.661 for the training set and 0.778 and 0.653 for the validation set. In the evaluations against the SEER-stage method, our nomograms consistently achieved a time-dependent AUC above 0.8, underscoring their enhanced forecasting process (Appendix A). The calibration curves displayed a close match between the forecasted and actual survival rates. The presented models precisely forecasted OS and CSS for all mentioned durations in both cohorts (Appendix A). Decision curve assessments for the OS and CSS models confirmed their elevated clinical relevance and forecasting competency for the stated durations, as illustrated by the expansive range of optimal threshold probabilities (Appendix A). Moreover, in the external validation cohort, metrics such as the calibration curve, time-dependent ROC, DCA curve, and risk stratification analysis unequivocally showed the robustness and superiority of the model (Appendix A).

### 3.6. Risk Stratification and Sankey Diagram Based on the Model

Using the Surv_miner R package, we established an optimal threshold to segregate patients into high-risk and low-risk categories concerning OS and CSS, with scores of 135 and 155, respectively. There was a pronounced divergence in survival trajectories among these risk groups (*p* < 0.001), underscoring the relevance of our nomogram and the stratification approach (Figure 5A–D). Furthermore, we leveraged heat maps to visualize variations in clinical features among the OS (Figure 5E) and CSS (Figure 5F) designated risk brackets. The model’s practical utility in clinical settings can be enhanced by illustrating a Sankey diagram that delineates the progression of each factor and its culmination into a designated risk category. As displayed in Figure 6A,B, this visual tool elucidates the influence of individual variables on the resulting risk classification.

### 3.7. Optimal Treatment Strategy Analysis

To examine how diverse treatments influence patient prognosis, propensity Score Matching analysis was applied to mitigate the influence of confounding factors [17]. The results for propensity score matching are presented in Appendix A. Prior to the matched assessment, triple therapy indicated more favorable OS and CSS outcomes than SR/SC. The comparative median survival periods were 10 months versus 91 months and 11 months versus 91 months. Notably, the five-year OS for triple therapy reached 57.1% and 58.7% for CSS, in contrast to SR/SC, which stood at 26.2% for OS and 29.3% for CSS (Figure 6C,D). Post-matching, the edge triple therapy persisted in terms of OS and CSS. The median survival intervals were 10 months juxtaposed at 93 months and 11 months compared with 93 months. Furthermore, the five-year SR/SC survival rates were 26.2% (OS) and 29.3% (CSS), while for triple therapy, they were 55.5% (OS) and 57.2% (CSS) (Figure 6E,F).

### 3.8. Genetic Mutations and GO/KEGG Analysis

The AT/RT genetic mutation data were extracted from COSMIC (https://cancer.sanger.ac.uk/cosmic, accessed on 30 December 2023) version GRCh38 COSMIC v99. In total, 17,650 cases of CNS tumors were evaluated for genetic mutations in the database. In the subselection category, all brain sites were selected for data extraction. For histological selection, only AT/RT cases were selected, and a final total of 194 cases were analyzed for genetic mutations. The top 20 genes that were mutated in AT/RT were SMARCB1 45% (in all samples tested = 269), BRAF 8% (73), SMARCA4 8% (26), NF2 2% (45), NRAS 2% (43), TP53 2% (43), KRAS 2% (43), MSH2 2% (43), IDH2 3% (31), KAT6B 4% (24), GATA2 4% (24), FANCD2 4% (24), SPEN 4% (24), ZFHX3 4% (24), NCOR2 4% (24), BCL11B 4% (24), ERBB4 4% (24), AFF3 4% (24), MAF 4% (24), and MDM4 4% (24) (Figure 7A). An overview of the mutation types and protein interactions network are shown in Figure 7B,C. We performed GO and KEGG analyses of these genes. Biological process analysis showed that the top 20 genes were enriched in RNA transcription, transcription factor complex, DNA-binding transcription factor, and ErbB signaling pathways (Figure 7D). Appendix A summarizes the details of the GO functions and KEGG pathways of the top 20 genes for co-expression enrichment analysis.

## 4. Discussion

AT/RT, recognized as a profoundly malignant CNS tumor, has a particularly poor prognosis, marked by an unusual level of aggressiveness [1,2,3,18]. Given the rarity of AT/RT instances, adequate research poses a challenge. The SEER database is a reputable repository of U.S. cancer statistics, making it instrumental in probing uncommon tumors [19,20]. Through analysis, we engaged a sizable cohort of AT/RT individuals from SEER (*n* = 316) and an external validation cohort from Chongqing, China (*n* = 27), subsequently discerning six clinical determinants linked to OS and CSS. Additionally, we used the COSMIC database to analyze the genomic variation characteristics of AT/RT. To the best of our knowledge, this inquiry marks a seminal SEER- and COSMIC-driven exploration to formulate predictive frameworks for specific survival to AT/RT. To amplify the real-world applicability of our findings, we incorporated web-enabled prognostic tools and devised a visual representation of Sankey to facilitate risk-based clinical decisions.

The actual incidence of AT/RT might have been underestimated owing to gaps in prior knowledge. Our epidemiological survey demonstrated a noticeable increase in the incidence of AT/RT over the past two decades. Consequently, it is imperative to prioritize and enhance AT/RT-related management in the future. Research indicates that AT/RT primarily targets infants younger than three years [3], with age being a pivotal determinant of patient outcomes. Our analysis confirmed that a significant fraction of the patient population aged <3 years (*n* = 248, 78.5%) demonstrated inferior prognostic outcomes. While tumor dimensions serve as a pivotal influencer on the outcomes of diverse solid tumors [21] by mirroring the tumor’s reach, our findings are consistent with clinical anecdotes, establishing that an expanded tumor size (≥4 cm) negatively affects both OS and CSS. However, a universally recognized staging forecast system for AT/RT remains elusive. The SEER database employs a novel approach that uses a combined summary stage [22]. This unique categorization delineates cancer progression from its inception into localized, regional, or distant stages, aiming to streamline clinical reference. Leveraging this SEER stage in our study, we found that a significant proportion of patients (*n* = 127, 40.2%) had already progressed beyond the localized phase upon initial identification.

For aggressive brain tumors, achieving maximal safe resection is the accepted standard for surgical management [23]. Our research further investigates how medical interventions tie with patient outcomes, underscoring the pivotal role that surgery holds in bolstering OS and CSS for AT/RT treatments. However, some studies have questioned the pronounced survival benefit of complete surgical intervention over other treatment modalities, suggesting that adjunct postoperative therapies might hold more weight [24,25]. The efficacy of high-dose chemotherapy in managing AT/RT has undergone rigorous examination, with prevalent regimens including the American COG’s CCG9933 and German GPOH’s HIT program [26,27]. Athale et al.’s meta-analysis postulated a potential survival advantage with intrathecal chemotherapy, especially for those unsuitable for craniospinal radiotherapy [28]. Furthermore, conjoining postoperative local radiotherapy with craniospinal radiotherapy demonstrates a remarkable potential to enhance local disease control and prolong survival [8]. Given the adverse neurotoxic side effects inherent to traditional radiotherapy and the looming threat of secondary malignancies, judicious consideration is imperative when choosing surgical strategies, orchestrating radiotherapy and chemotherapy sequences, and calibrating dosages for AT/RT patients under three years. Recent advancements have highlighted the potential of proton therapy to augment survival by fine-tuning radiation responses [29]. COG’s current investigation into AT/RT posits that radiation for children under three years should be primarily tumor-focused and recommends tapering the dosage for whole brain and spinal cord radiotherapy in metastatic cases from 30 Gy to 24 Gy [13]. Recognizing the ambiguity surrounding current treatments, our exploration gravitates towards triple therapy’s promise in extending survival durations. Propensity score matching evaluations indicated that, even after equalizing confounding elements between cohorts, the triple therapy recipients outperformed their counterparts (who underwent surgery accompanied by chemotherapy/radiotherapy) in OS and CSS. This finding is consistent with the results of previous studies and once again emphasizes the superiority of triple therapy but ignores side effects, such as radiotoxicity [30,31]. When it comes to childhood brain tumors, the most crucial factor to consider is the child’s brain. As highlighted by Walker’s research, conventional approaches that prioritize overall survival as the primary drive for change overlook the significant long-term significance of children’s brain health [32]. Hence, forward-focused clinical trials are essential to corroborate the efficacy and safety of this therapeutic approach.

However, the origin of AT/RT remains unclear. Based on the genetic and DNA methylation status and transcriptome profiles, AT/RTs are further divided into three distinct molecular subgroups: ATRT-SHH, ATRT-TYR, and ATRT-MYC [33]. Both the previous literature and our analysis suggest that mutations or absences in SMARCB1(INI-1) play a pivotal role in its development and progression [6,34]. INI-1 is ubiquitously expressed in the nucleus of normal cells and is considered a tumor suppressor gene. Studies have shown that INI-1 is a core component of the switch/sucrose-non-fermentable (SWI/SNF) chromatin remodeling complex, which regulates gene expression important for lineage specification and maintenance of stem cell pluripotency [35,36]. The GO/KEGG analysis based on the top 20 mutated gene sets also revealed that the biological properties of AT/RT are related to RNA transcription regulation, transcription factor complex formation, RNA-DNA specific binding, and the ErbB signaling pathway. Several studies showed that lapatinib has a good inhibitory effect on AT/RT by targeting the EGFR-ErbB2 signaling pathway [37,38]. Therefore, conducting an in-depth genomic analysis of AT/RT patients is necessary as it will provide potential therapeutic targets for this disease.

In the current investigation, the LASSO and RF regression methodologies were used to craft a model aimed at assessing survival threats and the genomic landscape used to clarify the underlying pathogenesis. Despite its strengths, our study has some limitations. Given the retrospective nature of the analysis, we bypassed patients who were absent from the SEER registry, potentially introducing a sampling bias. The SEER datasets also did not provide exhaustive details concerning pivotal clinical aspects, such as performance status, disability data, specific chemotherapy regimens, number of cycles, radiation dosages, and subsequent lines of therapy. The absence of metrics, such as disease progression-free survival and recurrence survival, in the SEER database could also pose limitations to the broader utilization of the model. Genomic analysis needs to be more in-depth; for example, an exploration of epigenetic changes and functional mechanisms could be performed.

## 5. Conclusions

We meticulously examined patient data from the SEER database spanning the years 2000 to 2019 and the genetic mutation characteristics of the patients in the COSMIC database. Our study identified the clinical determinants of prognosis in patients with AT/RT and mapped the genetic mutation landscape. The prediction model we have devised, characterized by its accuracy, might offer a valuable tool to address existing clinical challenges. Notably, the insights gleaned suggest the potential of triple therapy in refining patient prognosis. Additionally, analysis based on mutational genomics will facilitate research regarding molecular-targeted drugs. Collectively, our observations are intended to help clinicians better stratify patients for various treatments, measure the impact of treatments on demographic and tumor characteristics, and more accurately assess prognosis. As clinical understanding deepens and genomic profiling expands, precise and efficient combination therapies are expected to drive this disease in a positive direction.

## Figures and Tables

**Figure 1 cancers-16-01059-f001:**
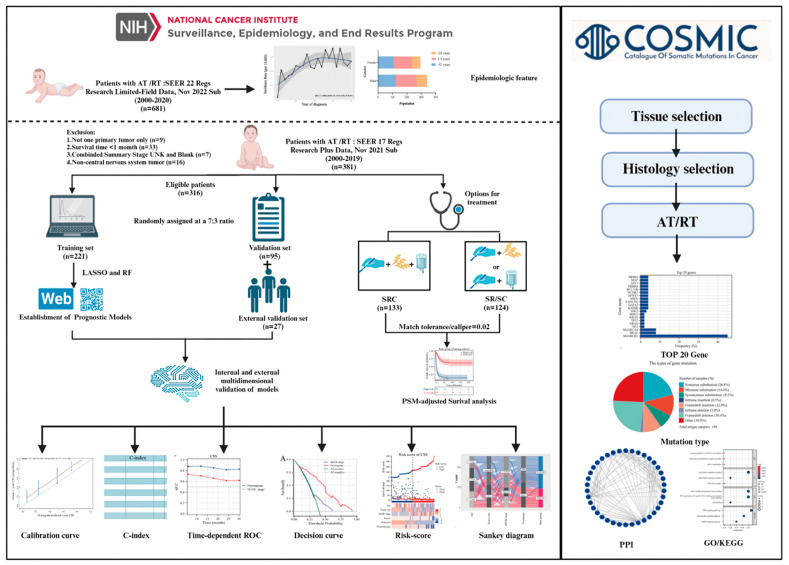
Study design and the workflow diagram.

**Figure 2 cancers-16-01059-f002:**
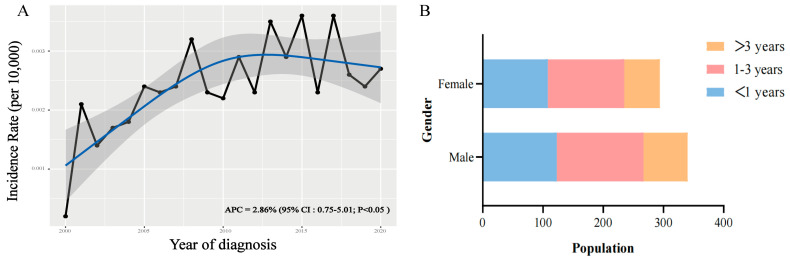
Trends and characteristics of AT/RT. (**A**) Incidence of AT/RT from 2000 to 2020. (**B**) Age and gender distribution of the total population from 2000 to 2020.

**Figure 3 cancers-16-01059-f003:**
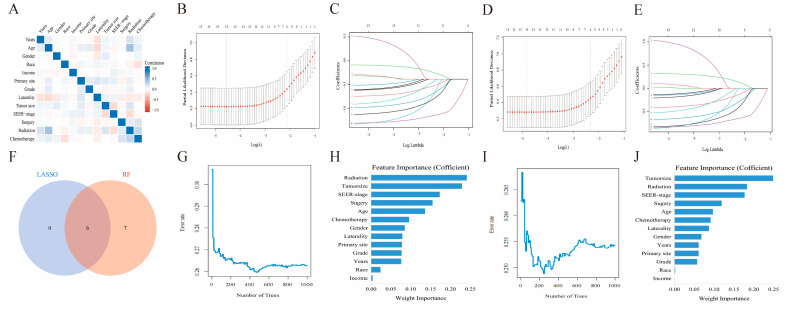
Correlation analysis, LASSO, and RF regression model in both OS and CSS. The results of correlation analysis between all included variables (**A**). Selection of tuning parameter (λ) for the LASSO model (**C**,**E**), 10-fold cross-validation (**B**,**D**). The out-of-bag error rate of Random forest (**G**,**I**) and variable importance ranking (**H**,**J**). Intersection variables of the two algorithms (**F**).

**Figure 4 cancers-16-01059-f004:**
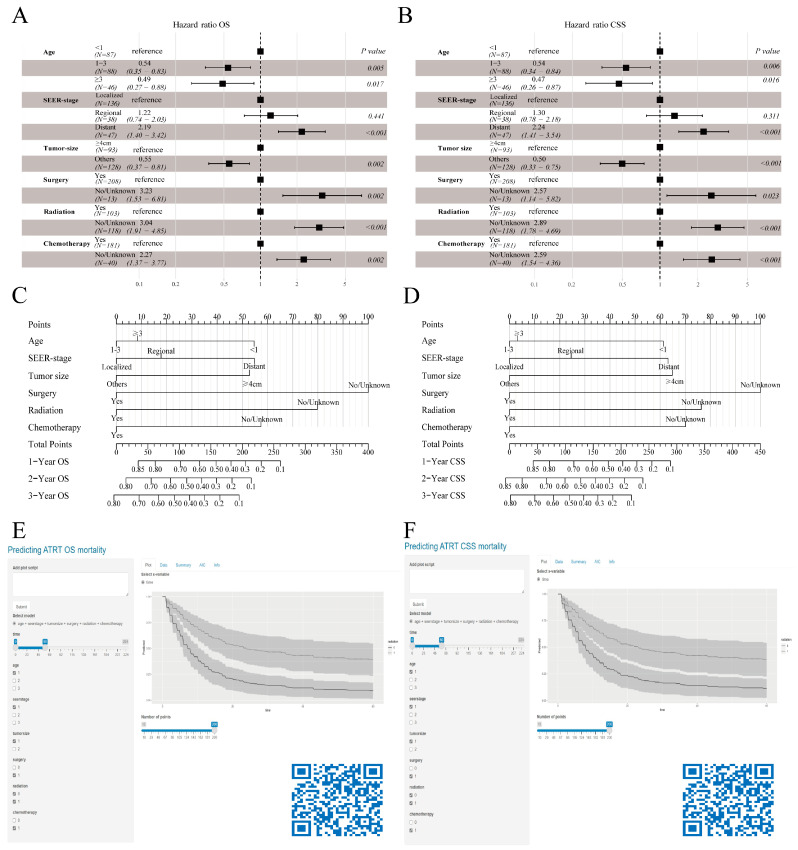
The forest plot and Nomogram. Forest plot demonstrating the model for OS (**A**) and CSS (**B**) in the training cohort and the Nomogram for predicting 1-, 2- and 3-year OS (**C**) and CSS (**D**). Dynamic survival risk system for OS (**E**) and CSS (**F**) in AT/RT patients.

**Figure 5 cancers-16-01059-f005:**
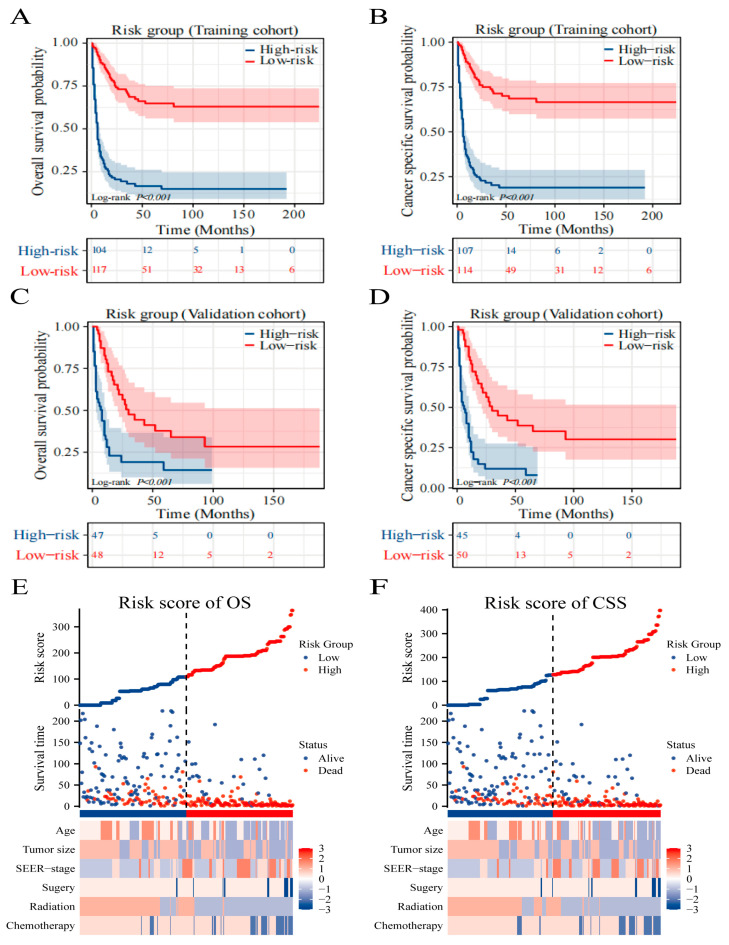
Overview of risk-stratification system according to risk points calculated by the model. OS and CSS analysis of patients with AT/RT in the training cohort (**A**,**B**) and validation cohort (**C**,**D**). The distribution of clinicopathological features in different risk groups for OS (**E**) and CSS (**F**).

**Figure 6 cancers-16-01059-f006:**
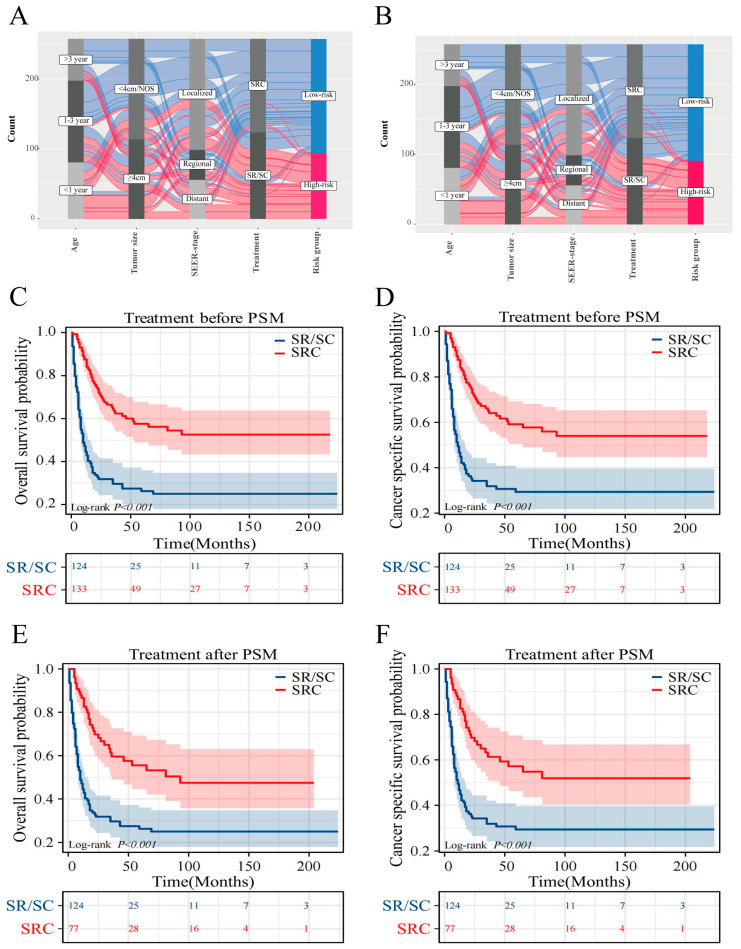
The Sankey diagram and Propensity score matching analysis. Sankey diagram of each predictor feature and risk grouping of OS (**A**) and CSS (**B**). The survival curves of two groups before and after matching analysis in OS (**C**,**E**) and CSS (**D**,**F**).

**Figure 7 cancers-16-01059-f007:**
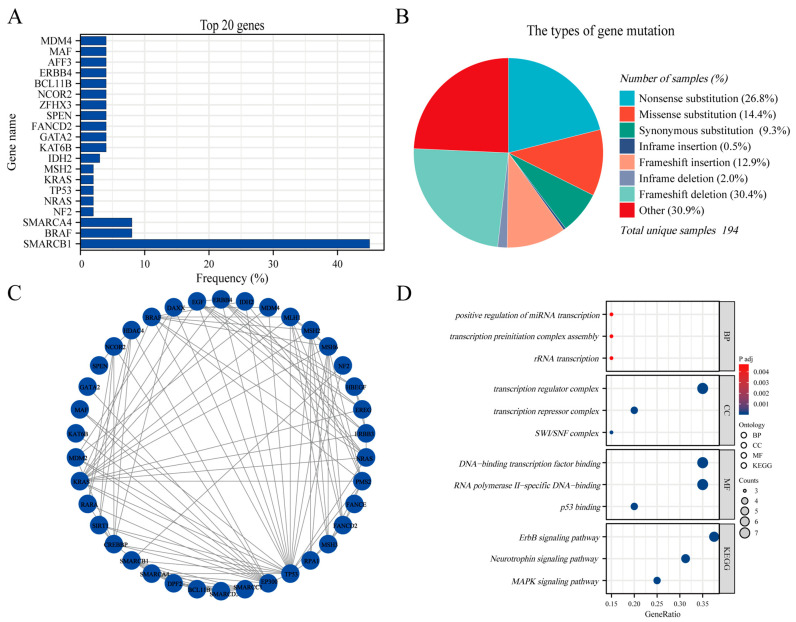
The Genomic Landscape, PPI and GO/KEGG analysis in AT/RT. (**A**) The top 20 mutated genes. (**B**) An overview of the types of mutation observed. (**C**) Protein interactions network of the top 20 mutated genes. (**D**) The GO/KEGG enrichment analysis of top 20 genes.

**Table 1 cancers-16-01059-t001:** Characteristics of patients with AT/RT in the training and validation group.

Characteristics	Total (*n* = 316)	Training Group (*n* = 221)	Validation Group (*n* = 95)	*p Value*
	No.(%)	No.(%)	No.(%)	
Years of diagnosis				0.361
2000–2009	137 (43.4%)	100 (45.2%)	37 (38.9%)	
2010–2019	179 (56.6%)	121 (54.8%)	58 (61.1%)	
Age				0.171
<1 year	114 (36.1%)	87 (39.4%)	27 (28.4%)	
1–3 year	134 (42.4%)	88 (39.8%)	46 (48.4%)	
>3 year	68 (21.5%)	46 (20.8%)	22 (23.2%)	
Gender				0.786
Male	165 (52.2%)	117 (52.9%)	48 (50.5%)	
Female	151 (47.8%)	104 (47.1%)	47 (49.5%)	
Race				0.295
White	241 (76.3%)	173 (78.3%)	68 (71.6%)	
Black	41 (13.0%)	28 (12.7%)	13 (13.7%)	
Others	34 (10.7%)	20 (9%)	14 (14.7%)	
Household income				0.987
<75,000$	221 (69.9%)	154 (69.7%)	67 (70.5%)	
≥75,000$	95 (30.1%)	67 (30.3%)	28 (29.5%)	
Grade				0.893
Unknown	253 (80.1%)	176 (79.6%)	77 (81.1%)	
III–IV	63 (19.9%)	45 (20.4%)	18 (18.9%)	
Primary site				0.194
Intracranial	299 (94.6%)	212 (95.9%)	87 (91.6%)	
Spinal cord	17 (5.4%)	9 (4.1%)	8 (8.4%)	
Laterality				0.674
Left	51 (16.1%)	35 (15.8%)	16 (16.8%)	
Right	55 (17.4%)	36 (16.3%)	19 (20%)	
Others	210 (66.5%)	150 (67.9%)	60 (63.2%)	
Tumor size				0.152
<4 cm/NOS	174 (55.1%)	128 (57.9%)	46 (48.4%)	
≥4 cm	142 (44.9%)	93 (42.1%)	49 (51.6%)	
SEER-stage				0.348
Localized	189 (59.8%)	136 (61.5%)	53 (55.8%)	
Regional	61 (19.3%)	38 (17.2%)	23 (24.2%)	
Distant	66 (20.9%)	47 (21.3%)	19 (20.0%)	
Surgery				0.999
GTR/STR	297 (94.0%)	208 (94.1%)	89 (93.7%)	
No/Unknown	19 (6.0%)	13 (5.9%)	6 (6.3%)	
Chemotherapy				0.647
Yes	256 (81.0%)	181 (81.9%)	75 (78.9%)	
No/Unknown	60 (19.0%)	40 (18.1%)	20 (21.1%)	
Radiation				0.999
Yes	148 (46.8%)	103 (46.6%)	45 (47.4%)	
No/Unknown	168 (53.2%)	118 (53.4%)	50 (52.6%)	

## Data Availability

All data generated or analyzed during this study are included in this published article and its additional information files. Further inquiries can be made at https://github.com/SihaoChen95/ATRT (accessed on 8 January 2024).

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
