# Peer review of "Dynamic Survival Risk Prognostic Model and Genomic Landscape for Atypical Teratoid/Rhabdoid Tumors: A Population-Based, Real-World Study"

_cancers, 2024, doi:10.3390/cancers16051059_

Round 1

Reviewer 1 Report

Comments and Suggestions for Authors

The manuscript is extremely interesting and the results make it clear that the objectives were achieved.

The authors described that atypical teratoid/rhabdoid tumor (AT/RT) is an uncommon but aggressive pediatric neoplasm of the central nervous system. The present prognostic study involved more than 300 people from the SEER database and 27 external validation patients.

Data demonstrated that the incidence of AT/RT increased consistently between 2000 and 2020. Age, SEER stage, tumor size, surgery, chemotherapy, and radiotherapy are closely related to the prognosis of AT/RT. Triple therapy resulted in noticeably improved OS and CSS.

The most common mutations in AT/RT identified in the COSMIC database were SMARCB1, BRAF, SMARCA4, NF2, and NRAS. This interesting study identified the clinical determinants of prognosis in patients with AT/RT and mapped the landscape of genetic mutations. The prediction model developed in this work can offer a valuable tool to address existing clinical challenges. Furthermore, analysis based on mutational genomics will facilitate the search for molecularly targeted drugs.

I consider the manuscript suitable for publication in the journal

Author Response

Thank you for taking an interest in our article and your thoughtful comments. It is our honor to get your recognition!

Reviewer 2 Report

Comments and Suggestions for Authors

This is a retrospective analysis using complex mathematical modelling of a SEER data set with an external clinical validation cohort of institution sourced patient data.  The clinical topic is the childhood brain tumour Atypical Teratoid Rhabdoid Tumour.  This has been selected for this complex analysis because of its rarity and challenging clinical characteristics of rapid growth and resistant phenotype.  Its unique genetic characteristics make it a strong candidate for molecularly targetable treatment once a drug and its optimal delivery method and target can be paired. The authors have gone to considerable lengths to try and establish a predictive model for survival.  The data sources they use do not include quality of life or disability data.  The details considering treatment are limited to what was recorded in the database.  The emphasis of the modelling is primarily overall survival and there is an assumptive thread in the writing that their model has the power to predict outcomes based upon patient, clinical and treatment variables.  In reality, in childhood cancers, prognostic variables based upon patient and tumour / biological characteristics determine sensitivity to treatment characteristics and so should be referred to as treatment selection criteria because without such treatments all patients would die and with treatments of different sorts some will survive. The analysis does examine the impact of treatment modalities used, specifically, surgery, drug therapy and radiotherapy.  However these are diverse and poorly characterised and there are gaps in knowledge of what happened to individual cases ie were treatments used as primary treatments or at relapse.  There are international efforts to study specific treatments and the impact of the genetic characteristics that are a prominent feature of this tumour type.  Overall, whilst this highly complex modelling process with its clinically sourced validation cohort  has good methodological and mathematical credentials it does not offer clinical trialists parameters they can use for trial design or discover new facts about the disease that have not been studied in other ways.  ATRT is a very rare disease, presents in early life and treatment selection is a balance of managing potentially neurotoxic therapies in young children's developing brains.   Any disability acquired has lifelong consequences for the child and family if the child survives.  Without information about disability or quality of life outcomes I do not feel this modelling exercise moves the scientific understanding of ATRT further forward (ref).  For this reason the impact of this work is compromised despite the high level of mathematical complexity.

Ref: Childhood brain tumors: It is the child’s brain that really matters David Walker Front. Oncol., 04 October 2022 Sec. Pediatric Oncology Volume 12 - 2022 | https://doi.org/10.3389/fonc.2022.982914

Author Response

Response: We thank the referees for helpful and constructive comments during the peer review process. The data included in this study do not provide details on patients' quality of life and specific treatments, which is a limitation of the study. We acknowledge this limitation and will address it in the discussion section. We will also include a discussion on the concerns about the quality of life of children and refer to relevant literature. Please see the line 359-363 of the revised manuscript: When it comes to childhood brain tumors, the most crucial factor to consider is the child's brain. As highlighted by David's research, conventional approaches that prioritize overall survival as the primary drive for change overlook the significant long-term significance of children's brain health (33).

Here, we would like to reiterate the significance of this study to gain your understanding. Our study focused on the survival length of children and aimed to identify clinical characteristics related to prognosis. Despite the lack of information on quality of life and specific treatments, the retrospective study successfully achieved its purpose. It is worth mentioning that the sample size of this study is sufficient, the model construction method is robust, and the internal and external validation is accurate. In the absence of a clinical prognosis assessment system, this model may assist frontline clinicians in determining treatment standards. Additionally, the study analyzed the genetic mutation characteristics of patients and may provide a reference for future development of molecularly targeted drugs. Given the rarity and high malignancy of this tumor, we believe that any relevant efforts and potential research are valuable for advancement.

Reviewer 3 Report

Comments and Suggestions for Authors

The aim of the submitted manuscript was to create a dynamic survival risk prognostic model for atypical teratoid/rhabdoid tumor. In this retrospective study the Authors have used proper methods and described them accurately. The results were properly discussed, and the conclusions, though limited, are correct. My major aim is about the quality of actually all of the figures. The font size used to create them is simply too small, to read them. For example, Figure 7C, even if you magnify it 250% it is still impossible to read the symbols.

Other comments are listed below.

Line 18, SEER acronym must be defined here

Line 24, „The prediction model that we”-this part is written using different font

Line 53, why is “Atypical” with capital “A”?

Line 59, please list those other embryonal tumors of the CNS

Line 70, “infrequency” is in contrast to what was written in lines 56-57

Line 95, a reference is needed here

Lime 151, I can’t agree with that. From Figure 1A it is visible that incidence increased only between 2000-2010 and stayed constant, or even slightly decreased, between 2010 and 2020.

Figure 2B, is it a cumulative data for 2000-2020?

Table 1, why was this particular household income (75000$) chosen as a value?

Author Response

Thank you for taking an interest in our article and for your thoughtful comments. Below is our point-by-point response to your comments.

  1. Figure 7C, even if you magnify it 250% it is still impossible to read the symbols.

Response: We have resized the font and made it bold. Details can be found in the uploaded revised draft.

  1. Line 18, SEER acronym must be defined here.

Response: We have corrected the abbreviation here and marked it in red with the corresponding font. Details can be found in the uploaded revised draft.

  1. Line 24, “The prediction model that we”-this part is written using different font

Response: We have reworked the font size to ensure uniformity, thank you very much for pointing this out.  Details can be found in the uploaded revised draft.

  1. Line 53, why is “Atypical” with capital “A”?

Response: We thank the reviewer for pointing this out! This was a mistake on our part and we have since corrected it. Details can be found in the uploaded revised draft.

  1. Line 59, please list those other embryonal tumors of the CNS.

Response: We thank the reviewer for pointing this out! We have added relevant descriptions to the corresponding locations in the manuscript and displayed them in red font. Details can be found in the uploaded revised draft.

  1. Line 70, “infrequency” is in contrast to what was written in lines 56-57.

Response: Thank you very much for your advice and comments! In fact, rare here refers to rare among all cancers. To avoid misunderstandings, we have stated this accordingly in the manuscript. Please see line 71 of the revised manuscript: The infrequency of AT/RT in the general population and its diagnostic intricacies mean that prior investigations of this ailment have largely focused on individual case analyses and modest retrospective evaluations.

  1. Line 95, a reference is needed here

Response: We thank the reviewer for pointing this out! We have added references at appropriate locations in the manuscript. Details can be found in the uploaded revised draft.

[Reference: Collins GS, Reitsma JB, Altman DG, Moons KG. Transparent reporting of a multivariable prediction model for individual prognosis or diagnosis (TRIPOD): the TRIPOD statement. BMJ. 2015 Jan 7;350:g7594. doi: 10.1136/bmj.g7594.]

  1. Lime 151, I can’t agree with that. From Figure 1A it is visible that incidence increased only between 2000-2010 and stayed constant, or even slightly decreased, between 2010 and 2020. Figure 2B, is it a cumulative data for 2000-2020?

Response: Thank you very much for your advice and comments! Actually, the "increase" here refers to the overall incidence rate increasing over the past 20 years. Indeed, the incidence rate seems to have declined slightly in 2010-2020, but overall it has increased compared with the initial period (APC=2.86%, P < 0.05). Sorry to disturb your understanding, Figure 2B shows the accumulated quantity over 20 years. We have clarified this in the revised manuscript. Details can be found in the uploaded revised draft.

  1. Table 1, why was this particular household income (75000$) chosen as a value?

Response: Thank you very much for your advice and comments! There are two main considerations in choosing this cutoff point. First, $75,000 is the maximum value recorded in the database; second, we focus on whether differences in family economic conditions have an impact on patient prognosis. To sum up, we chose the maximum cutoff point.

Reviewer 4 Report

Comments and Suggestions for Authors

This research tries to identify the clinical determinants of prognosis in patients with atypical teratoid/rhabdoid tumor and mapped the genetic mutation landscape. The supplied prediction model is supposed to offer a valuable tool to address existing clinical challenges. Analysis based on mutational genomics will facilitate the research of molecular targeted drugs. Overall, this manuscript was prepared with sufficient amount of data and discussions. It may be publishable in Cancers with necessary revisions. Rather than actual scientific data, this manuscript needs revisions on general points. Especially readability has to be well improved.

1) Although summary of abbreviations is listed at the last, I feel difficulty  in reading this manuscript with many abbreviations everywhere. Some easy words are better to be non-abbreviated. At least, the abbreviation table had better be listed at the beginning of the manuscript.

2) In many figures, inside words cannot be seen well. Many panels are integrated in one figure, and each panel becomes very small because of bad arrangement. Please seriously improve this point.

3) Although this manuscript is written well. conclusions is rather insufficient. More descriptions on research impacts and future perspectives had better be added.

Author Response

Thank you for taking an interest in our article and for your thoughtful comments. Below is our point-by-point response to your comments.

  1. Although summary of abbreviations is listed at the last, I feel difficulty in reading this manuscript with many abbreviations everywhere. Some easy words are better to be non-abbreviated. At least, the abbreviation table had better be listed at the beginning of the manuscript.

Response: We thank the reviewer for pointing this out! There are indeed a large number of abbreviations in this article, which will cause some discomfort to readers. We have removed some simple abbreviations from the manuscript, such as PPI, PSM. And we have agreed to move the abbreviation order forward. However, since the abbreviation list in the Cancers format template is located at the end of the article, we will contact the editor to seek clarification on whether the order can be changed. Details can be found in the uploaded revised draft, and the modifications are also shown in red font in the manuscript.

  1. In many figures, inside words cannot be seen well. Many panels are integrated in one figure, and each panel becomes very small because of bad arrangement. Please seriously improve this point.

Response: Thank you very much for your advice and comments! We acknowledge your observation regarding the inappropriate fonts and arrangement of figures in the manuscript. In response, we have addressed this concern by enlarging the font in the image and uploading a clearer original image. Details can be found in the uploaded revised draft.

  1. Although this manuscript is written well. conclusions is rather insufficient. More descriptions on research impacts and future perspectives had better be added.

Response: Thank you very much for your advice and comments! We strongly agree with your point of adding the impact of the study and future perspectives in the conclusion. Please see the lines 393-400 of the revised manuscript: The prediction model we've devised, characterized by its accuracy, might offer a valuable tool to address existing clinical challenges. Notably, the insights gleaned suggest the potential of triple-therapy in refining patient prognosis. Additionally, analysis based on mutational genomics will facilitate the research of molecular targeted drugs. Collectively, our observations are intended to help clinicians better stratify patients for various treatments, measure the impact of treatments on demographic and tumor characteristics, and more accurately assess prognosis. Details can be found in the uploaded revised draft, and the modifications are also shown in red font in the manuscript.

Round 2

Reviewer 2 Report

Comments and Suggestions for Authors

The revision has addressed some aspects of the review. They cite the opinion paper with my first name rather then the conventional surname. It is the editors decision as to whether the journal should publish this. The mathematical modelling is complex but omits QoL data and is unable to identify specific therapies at specific ages to be of clinical use as claimed. The main scientific value is as a power calculation for the use of this type of analysis and ideally should be reported as such especially if QoL outcomes were considered

Author Response

Dear reviewer,

Thank you for taking an interest in our article and for your thoughtful comments. We acknowledge that this study has certain limitations as you mentioned and have corrected the author names of the cited references in the latest revised version. Details can be found in the uploaded revised draft, and the modifications are also shown in red font in the manuscript.

Reviewer 3 Report

Comments and Suggestions for Authors

The Authors have answered my questions and done the required corrections.

Author Response

(The authors gave the same response as above.)
